# Antibacterial Properties of Small-Size Peptide Derived from Penetratin against Oral Streptococci

**DOI:** 10.3390/ma14112730

**Published:** 2021-05-21

**Authors:** Meng Li, Yanyan Yang, Chen Lin, Qian Zhang, Lei Gong, Yonglan Wang, Xi Zhang

**Affiliations:** 1School and Hospital of Stomatology, Tianjin Medical University, 12 Observatory Road, Tianjin 300070, China; mengli8344@tmu.edu.cn (M.L.); yyy@tmu.edu.cn (Y.Y.); linchen666@tmu.edu.cn (C.L.); zhangqian1218@tmu.edu.cn (Q.Z.); 2Affiliated Stomatology Hospital of Guangzhou Medical University, Guangzhou 510150, China; 3Fengtai Maternal & Child Health Hospital, Beijing 100069, China; 4Department of Esophageal Cancer, Tianjin’s Clinical Research Center for Cancer and Key Laboratory of Cancer Prevention and Therapy, National Clinical Research Center for Cancer, Tianjin Medical University Cancer Institute and Hospital, Tianjin 300070, China; leigong@tmu.edu.cn

**Keywords:** small peptides, RR9, oral biofilms, human gingival fibroblasts, anti-inflammatory activity

## Abstract

Periodontitis, an infectious disease originating from dental biofilms that causes the irreversible loss of alveolar bone, is accompanied by gradual biofilm formation and the continuous progression of inflammation. A small peptide derived from penetratin, Arg-Gln-Ile-Arg-Arg-Trp-Trp-Gln-Arg-NH_2_ (RR9), appears to have antibacterial properties against selected strains associated with periodontitis. The purpose of this research is to assess the antibacterial activity and mechanism of RR9 against the initial oral colonizers *Streptococci oralis*, *Streptococci gordonii*, and *Streptococci sanguinis* and to investigate the cytotoxicity of RR9 on human gingival fibroblasts in vitro. The effects of RR9 on the initial oral settlers of planktonic and biofilm states were evaluated by measuring the MIC, MBC, bactericidal kinetics, and antibiofilm activity. Visual evidence and antibacterial mechanisms were identified, and the anti-inflammatory activity and cytotoxicity were measured. The results demonstrated that RR9 can inhibit the growth of streptococci in the planktonic state and during biofilm formation in vitro while keeping a low toxicity against eukaryotic cells. The antibacterial mechanism was proven to be related to the lower expression of *sspA* in streptococci. RR9 may be used as a potential antimicrobial and anti-infective agent for periodontal disease.

## 1. Introduction

As a complex infectious disease induced by pathogenic microorganisms, periodontitis results in the immuno-inflammatory destruction of periodontal tissues. The subsequent tooth loss possibly occurs as a result of the decreased periodontal ligament and resorbed alveolar bone [1]. Periodontitis is the sixth most ubiquitous chronic disease, affecting more than 700 million people all over the world, with undesirable impacts on people’s oral and general well-being [2,3]. In addition, periodontal disease exhibits a complicated relationship with other systemic diseases, such as diabetes mellitus, cardiovascular diseases, etc. [4,5,6,7]. Plaque biofilm is the initiating factor of periodontal diseases [8]. Oral biofilms are communities with complex three-dimensional structures consisting of a broad range of multi-species microorganisms formed on colonizable surfaces [9]. Biofilm is one of the main forms of oral microorganisms’ survival, reproduction, and pathogenicity, and it is very difficult to be completely removed because of its structural complexity [9,10]. The colonization of biofilm on oral surfaces has been considered the leading cause of various infectious diseases in different fields of dentistry, including cariology, endodontics, and periodontics [10]. Among early-stage colonizers, *Streptococci gordonii*, *S. sanguinis,* and *S. oralis* have been responsible for the initial attachment during biofilm formation [11], and they are the basis of subsequent bacterial adherence.

The traditional non-surgical treatment for periodontitis involves the removal of plaque by periodontal scaling and root planing. This is followed by surgical intervention [12]. However, periodontal pathogens cannot be completely removed, whether it is through the classical non-surgical method of root planing or the surgical flap treatment [13]. Some patients have persistent infections and tissue destruction, which eventually lead to tooth loss [14]. In such cases, compared with mechanical debridement alone, such treatments in combination with other drugs may have better clinical efficacy. Due to the side effects of antibiotic use, there is a great need for original biological agents to treat and prevent the disease. With the development of an understanding of the working mechanism of natural cationic peptides, a few synthetic peptides have been prepared, and their potential effectiveness has been demonstrated via clinical testing trials [15]. However, despite high expectations for this approach, the high manufacturing costs to produce synthetic peptides cannot be neglected [16].

To overcome the drawbacks of using cationic peptides as a new generation of antibiotics, researchers began to synthesize new short peptides by reducing the residue of the peptide chain for use as candidate drugs in clinical trials. Recently, Arg-Gln-Ile-Arg-Arg-Trp-Trp-Gln-Arg-NH_2_ (RR9) has been shown to have a strong antimicrobial effect against selected bacteria [17]; however, whether the short peptide prevented biofilm formation was unclear. Therefore, we aimed to figure out the antibacterial activity of RR9 on initial oral colonizers, specifically *S. oralis*, *S. gordonii,* and *S. sanguinis*. We also analyzed the secondary structure of RR9 and investigated the influence of the peptides on the initial stage of oral biofilm formation. Additionally, we investigated the toxic effects of RR9 on the viability of human gingival fibroblasts (HGFs). This study aims to identify whether RR9 can interfere with biofilm formation and can prevent periodontal diseases.

## 2. Materials and Methods

### 2.1. Peptide Prepare

The small peptide RR9 was synthesized by Dechi Biosciences (Shanghai, China). The final chimeric peptide (over 95% purity) was purified by HPLC and characterized by mass spectrometry. The peptide powder was dissolved in 0.5% glacial acetic acid, and the solution was prepared using the original peptide solution and subsequent diluent for experiments. For the human gingival fibroblast (HGF) cell test, the peptide was dissolved in a serum-free DMEM medium. The peptide property calculator was used to forecast the basic properties of RR9 (http://www.pepcalc.com/, accessed on 10 February 2019).

### 2.2. Bacterial Growth

The bacterial strains *S. oralis* (ATCC No 9811), *S. sanguinis* (ATCC No 10556), and *S. gordonii* (ATCC No 10558) were purchased from the ATCC (American Type Culture Collection, Manassas, VA, USA). The cultivation method was the same as that described in a previous article [18].

### 2.3. Cell Cultures

HGFs were purchased from Procell Life Science & Technology (Wuhan, China). Cells from passages 4 to 6 were used. The cultivation method was described previously [19].

### 2.4. Circular Dichroism (CD) Spectroscopy

The RR9 solution was dissolved in a sterile PBS solution to make its final concentration 0.1 mg mL^−1^. CD measurements were carried out at room temperature by a BIO-LOGIC CD spectrometer (J-810, Seyssinet-Pariset, France). The results within 190–350 nm were obtained using solvent subtraction with a scanning speed of 60 nm min^−1^. The experimental results involved in this study were repeated in at least three independent experiments.

### 2.5. Antibacterial Activity Test

#### 2.5.1. Minimum Inhibitory Concentration (MIC) and Minimum Bactericidal Concentration (MBC)

The bacterial growth inhibiting ability of the peptide was determined using a broth microdilution-based method with modifications [20]. The MIC and MBC were tested toward *S. oralis*, *S. gordonii,* and *S. sanguinis* in triplicate in the concentration range of 25–125 μg mL^−1^. The start inoculum was 5 × 10^7^ CFU mL^−1^, and the MIC was denoted as the lowest concentration of peptides where bacterial growth was stopped. After bacteria were treated with RR9 for 24 h, 20 uL was removed from each sample and spread on a brain heart infusion (BHI) agar plate. The CFU on plates were counted after incubation at 37 °C for 5 days. The MBC is the concentration of RR9 where over 99.9% of the cells were inhibited.

#### 2.5.2. Biofilm Susceptibility

The antibacterial agents were evaluated by the microdilution method to study their impact on biofilm formation [21]. In our study, streptococci (5 × 10^7^ CFU mL^−1^) mixed with RR9 were cultured for 24 h. The formed biofilm was fixed using methanol (95%) and stained using 0.5% (*w*/*v*) crystal violet, and the final OD at 600 nm was measured.

### 2.6. Time-Kill Assay

To evaluate the effect of RR9 on planktonic oral streptococci, the time-killing assay was used [22]. First, 200 μL of RR9 solution and 800 μL of BHI broth were mixed with 1 mL of bacterial culture (5 × 10^7^ CFU mL^−1^), yielding 2 mL of the final volume with 100%, 200%, or 400% of the MBC final peptide concentrations. After 0, 0.25, 0.5, 1, 2, 4, 24, and 48 h of incubation at 37 °C, 10 μL of each suspension was pipetted, serially diluted 10 times, and finally transferred to a BHI agar plate for CFU determination. The time–kill kinetic curves were obtained by recording log(CFU mL^−1^) as a function of incubation time over 6 h.

### 2.7. Confocal Laser Scanning Microscopy (CLSM)

Each bacteria suspension (300 μL) was mixed with BHI broth (900 μL) and then cultured for 24 h in a 24-well microtiter plate for biofilm formation. After the removal of the medium and unattached bacterial cells, RR9 (25 μg mL^−1^) with saliva or serum was added, and the resulting samples were cultured for another 24 h at 37 °C. After the staining of the attached bacteria using an acridine orange/ethidium bromide (AO/EB) solution (Solarbio, Beijing, China), CLSM (Leica SP8, Oskar-Barnack-Strae Germany) was used to count the number of observed bacterial cells. For each biofilm strain, three representative areas per lens were scanned in at least three separate experiments. The images were captured using MetaMorph software (Universal Imaging, West Chester, PA, USA).

### 2.8. Electronic Microscopy Analysis

The morphology of the formed biofilm was observed using scanning electron microscopy (SEM, SUPRA 55VP, Germany) following previously published procedures [18]. Additionally, the transmission electron microscopy (TEM, JEM-2100F; JEOL Ltd., Tokyo, Japan)-based technique was employed for the structural analysis of RR9 cells according to the previously reported method [23]. For more details, please check Appendix A.

### 2.9. Real-Time Polymerase Chain Reaction (RT-PCR)

Trizol reagent was employed to extract the total RNA of *S. gordonii* cells, following the manufacturer’s instructions. The first strand of cDNA was synthesized by the retrotranscription of 2 μL of RNA (Invitrogen, TRIzol Max Bacterial RNA Isolation Kit). The cDNA was amplified to a final volume of 20 μL by 45 cycles in a DNA thermal circulatory apparatus. Pre-denaturation was carried out at 95 °C for 15 s, and the reaction was performed at 95 °C for 10 s, 60 °C for 20 s, and 72 °C for 30 s. The primer sequences of the differentiation markers are shown in Table A1 in Appendix B. After amplification, *GAPDH* was defined as the internal reference gene, and the sample of the control group was set as the standard to obtain the CT value of each target gene for each sample. The data were analyzed using the relative quantification of ΔCT. The relative quantitative (RQ) values of all target genes were obtained according to the formula RQ = 2^−ΔΔCT^.

### 2.10. Cell Counting Kit-8 Assay

The HGFs (2 × 10^3^ cells per well) were cultured in DMEM supplemented with 10% FBS in a humidified incubator (37 °C, 5% CO_2_ atmosphere) for 24 h. Pretreatment of the cells was conducted by incubating with 25 μg mL^−1^ RR9 for 1, 2, 3, and 4 days. Untreated cells were used as a negative control. The samples were characterized using the Cell Counting Kit-8 (Dojindo Laboratories, Kumamoto, Japan) by measuring their absorbance at 450 nm using a microplate reader (BioTek Synergy HT, Vermont, USA). After staining using AO/EB live/dead dye, the viability of seeded cells on the sterile coverslips was determined using CLSM. Each sample consisted of five wells.

### 2.11. Statistical Analysis

All experiments in the current study were carried out in triplicate or repeated at least three times. All data were depicted into the mean ± standard deviation (SD). ANOVA and independent sample *t*-tests were employed to identify the significant differences between groups (SPSS 20.0, IBM, Chicago, IL, USA). A value of *p* < 0.05 was denoted as statistically significant.

## 3. Results

### 3.1. Properties of Peptides

The sequence and structure of RR9 are shown in Figure 1A. According to the prediction results of the software, this peptide is a short peptide composed of nine amino acid sequences. It has a linear α-helical structure that is highly positively charged (+4), amphiphilic, and has a molecular weight of 1384.59 Da in an aqueous medium (Figure 1a). The secondary structure of the peptide is related to its properties, so we used CD spectroscopy to detect the structure (Figure 1B). As for the qualitative secondary structure assignments, the minimum signal was located at 218 nm while the maximum one was at 195 nm. Alternatively, no positive peak was assigned to any random coil with a minimum signal at 198 nm [23,24]. None of the CD signals of RR9 matched the above peaks, indicating that RR9 is unstructured.

### 3.2. Antibacterial Assays

The antibacterial activity of RR9 was evaluated using the microdilution method against streptococcus biofilm formation in vitro. The MIC and MBC values are listed in Table 1. The results indicate that the peptide has potential antibacterial activity against the three selected primary colonizers, i.e., *S. gordonii*, *S. sanguinis*, and *S. oralis*. Notably, RR9 exhibited a prominent selective antibacterial effect against *S. gordonii*, with the lowest MIC compared with the other streptococci, which showed that *S. gordonii* is more susceptible to RR9 (MIC and MBC counting for 25 and 50 μg mL^−1^, respectively).

The antimicrobial activity kinetics of RR9 were compared against the three bacterial strains, which showed that RR9 has a time-dependent bactericidal effect on streptococci, as the number of viable bacteria significantly decreased with increased action time. By increasing the concentration of RR9, the reduction in viable bacteria was more rapid. Bacterial death first occurred within 4 h of RR9 addition, and complete bacterial death was observed within 24–48 h in all tested strains (Figure 2).

### 3.3. Anti-Biofilm Activity

The biofilm susceptibility test showed that a reduced OD value was observed from the bacterial biofilms treated with RR9 compared with the control group, and their anti-biofilm properties were significantly different (*p* < 0.05). RR9 exhibited different anti-biofilm properties against the three different strains of streptococci, though the differences were statistically insignificant (*p* > 0.05) (Figure 3a). The green and red spots in the CLSM images represent the living and dead bacteria, respectively. In the control group, the biofilm of streptococci was dense and mainly green in the CLSM image. However, in the RR9 groups with saliva or serum, the biofilm of streptococci was loose, with a decreased number of living bacteria but increased number of dead bacteria. No significant difference was observed when comparing the RR9 group with the RR9 groups with saliva or serum (Figure 3b).

### 3.4. Direct Observation of Antimicrobial Activity

SEM and TEM were used to observe the bacterial morphology and biofilm formation. SEM images demonstrate the effect of RR9 treatment on the streptococci biofilms for 24 h (Figure 4A). The bacteria without RR9 treatment had smooth and complete surface morphologies with no obvious holes, cell shrinkage, bacterial swelling, cell lysis, or fragments on the surfaces of the cell membranes. In contrast, the streptococci biofilm treated with RR9 showed a decrease in biomass and a change in bacterial morphology. After RR9 treatment, the bacteria swelled, and some cell membranes were damaged. These results indicate that RR9 inhibited the formation of streptococci biofilms by affecting the permeability of the membrane. Compared with the control group, the integrity of the cell membranes in the experimental group were compromised, and there was obvious content overflow (Figure 4B). These results indicate the critical impact of RR9 in bacteriostasis or sterilization by affecting the bacterial cell membrane.

### 3.5. RR9 Downregulated the Expression of SspA

To verify the inhibition effect of RR9 on the expression of *sspA* genes and *S. gordonii*, RT-PCR experiments were carried out to determine the adhesion to the tooth surface and the subsequent colonization of bacteria (Figure 4C). In comparison with the control group, the expression of the *sspA* gene was significantly reduced (*p* < 0.05) in the presence of RR9. Thus, we can conclude that RR9 inhibits the expression of the *sspA* gene of *S. gordonii*.

### 3.6. Cytotoxic Activities of RR9-Treated HGFs

After 1 day of incubation, the number of HGFs in the experimental and control groups was low and the cells were in long fusiform. After 2 days, the number of cells in the two groups increased significantly and a few cells were connected. The number of cells in the experimental and control groups further increased, and the connection between cells was easier to observe (Figure 5a) after 4 days. No significant difference was obtained by comparing the OD 450 values between the two groups (*p* < 0.05). The results of AO/EB fluorescence staining and CLSM (Figure 5b) showed that, in the RR9 group, the living cells (green fluorescence) covered the whole surface of the coverslip in a long fusiform with few dead cells (red fluorescence) attached, and the number of cells gradually increased with culture time. Therefore, the negligible cytotoxicity and good biocompatibility of RR9 (25 μg mL^−1^) are ascertained.

## 4. Discussion

Periodontitis is an inflammatory disease caused by microorganisms that can lead to alveolar bone absorption and subsequent tooth loss. Oral streptococci are the early and late colonizers of plaque biofilm growth. An in vitro study showed that, once *S. gordonii* forms a biofilm on the tooth surface, it provides an adhesion matrix for *Porphyromonas gingivalis*. There are highly conserved antigen I/II family proteins on the surface of all streptococci in the human mouth. For *S. gordonii*, SspA and SspB proteins are representatives of the antigenic I/II family, which are often expressed on the cell wall and can be compared with the receptors in saliva or serum [25,26]. The adhesion of *P. gingivalis* and *S. gordonii* plays an important role in the formation of pathogenic plaque [27]. Among them, the members of the adhesin family of streptococcal antigen I/II are bound to *P. gingivalis* by the SspA and SspB proteins expressed by *S. gordonii* [28]. By interfering with the initial attachment of bacteria, biofilm formation can be hindered toward disease prevention and treatment.

Antibiotics are used excessively and inappropriately in clinic. As a result, it directly leads to the weakening of the effect of antibiotics and the emergence of drug-resistant pathogens [29,30]. Among new biomedical alternatives, antimicrobial peptides (AMPs) [31] and cell-penetrating peptides (CPPs) [32] are considered promising candidates. CPPs are a new carrier that can pass through the cell membrane without cell surface receptor recognition and can deliver therapeutic drugs to the cell membrane [33,34]. One of the most well-known examples of CPPs is penetratin (RQIKIWFQNRRMKWKK-NH_2_). A previous study showed that penetratin has a strong bactericidal effect on the tested bacteria without any cytotoxicity toward mammalian cells [35,36]. The small peptide RR9 (RQIRRWWQR-NH_2_) derived from penetratin is shorter and easier to synthesize but needs to be further evaluated in terms of its antimicrobial activity against oral colonizers.

In addition, the comprehensive characterization, application of secondary structure prediction, and structural analysis using the PSIPRED server and CD spectroscopy have indicated that RR9 contains four cationic, two polar, and three hydrophobic residues. RR9 was mostly unstructured in solution. Notably, the 208 and 222 nm were not its CD minima, and the maximum was not at 190 nm but slightly shifted to 195 nm (Figure 1b). The RR9 structure was not completely defined from individual CD data; however, it can be said that it is structurally flexible. Previously, researchers plotted Edmundson wheel representations for RR9, which displayed charged and noncharged residues. As an essential structural characteristic, this type of distribution is potentially representative of their antibacterial and antifungal activities [36]. The attraction between the opposite charges on lipids and peptides seems to affect the process of peptide membrane contact [37,38]. There are four R residues in RR9. The results show that the electrostatic interaction between the positively charged arginine and negatively charged phospholipid is the key to peptide/lipid recognition. In spite of the existing lipids, the peptides’ structure is not directly related to their biological characteristics. Interestingly, some peptides adopt an amphiphilic helix structure with negatively charged lipids [39]. Our study showed that RR9 has a destructive effect on bacterial cell membranes, which shows that it changes its structure in the presence of a phospholipid layer.

MIC, MBC, and anti-biofilm test results showed that RR9 has a strong antibacterial effect on streptococci, although there was no significant difference between the effects on the three types, indicating that RR9 has a similar inhibiting effect on the selected streptococcus. The lower MIC and MBC reduce the clinical application cost of antibacterial polypeptides. The germicidal kinetic curve showed that the germicidal efficacy of RR9 is dependent on time and concentration. The antimicrobial effect was first observed 4 h after the test began and lasted 44 h.

Currently, the mechanism of the antibacterial peptide membranes remains controversial. There are two ways for CPPs to affect bacterial cells: by causing damage to the cell membrane and thus increasing the cell permeability, or by entering the bacterial cell cytoplasm without damaging the cell membrane [40]. Through SEM and TEM, we can directly observe the destruction of the target cell ultrastructure by antibacterial peptides as well as the target of antibacterial action and the mechanism of antibacterial activity. Our results confirmed that RR9 destroys the plasma membrane and induces pore-like channels, resulting in membrane disruption and the leakage of cell contents (Figure 4a). SEM imaging (Figure 4b) also showed that RR9 inhibits the formation of *S. gordonii* biofilms, which is consistent with the antibacterial test results. RR9 destroys the envelope structure of the bacteria because the combination of RR9 and the cell membrane changes the nature of the latter. As such, SEM and TEM analyses were used to identify whether RR9 could lower the integrity of *S. gordonii* membranes.

When evaluating the possibility of drug applications, it is necessary to assess its cytotoxicity. An ideal periodontal treatment drug that not only has a good bacteriostatic effect but also protects periodontal tissue has a low toxicity [41]. In this study, the potential cytotoxicity of RR9 toward HGFs was evaluated. The results in this study show that RR9 is harmless toward HGFs (Figure 5). The cells in the experimental group and the control group grew with normal morphology and reproduction rate. In addition, RR9 exhibited no cytotoxicity toward HGFs, as evidenced by the relative viability and proliferation to control groups.

The healing time of gingival epithelial cells is 24 h after sub-gingival scaling and root planing [42]. According to the data, RR9 began to show antimicrobial activity after 4 h of contact with the bacteria, the effects of which last for 44 h. This indicates that RR9 had potential antimicrobial activity against streptococci and anti-inflammatory activity in the tissue healing process, which makes it a suitable agent to decrease the occurrence of periodontal disease. On the other hand, RR9 could reach the root surface of teeth before other bacteria. It then prevents biofilm formation by occupying the bacterial adhesion sites on the tooth surface after periodontal treatment.

## 5. Conclusions

Due to the side effects of antibiotics, the simultaneous use of low concentrations of drugs in adjuvant therapies may induce fewer adverse effects during periodontitis treatment. Cationic antimicrobial peptides have a wide range of antibacterial properties, among which the short peptide RR9 has a strong permeability and obvious advantages. However, its role in oral microecology remains unclear. We conclude that the small peptide RR9 has antibacterial abilities against oral streptococci and is stable in saliva and calf serum. RR9 also exhibits stable killing kinetics against oral streptococci and strong inhibitory activity toward biofilm formation. It was confirmed by visual observations that RR9 can penetrate plasma membranes, leading to oral streptococci death and biofilm disintegration. We also showed that RR9 has negligible cytotoxicity toward HGFs in vitro. Therefore, RR9 may have the potential for application as an antimicrobial and anti-infective agent for periodontal disease. There are some limitations to this study. To further explore the characteristics of RR9 and its potential application value, the peptide must be used in a more realistic and complex biofilm model, and its anti-inflammatory mechanism in periodontitis must be further explored in vivo.

## Figures and Tables

**Figure 1 materials-14-02730-f001:**
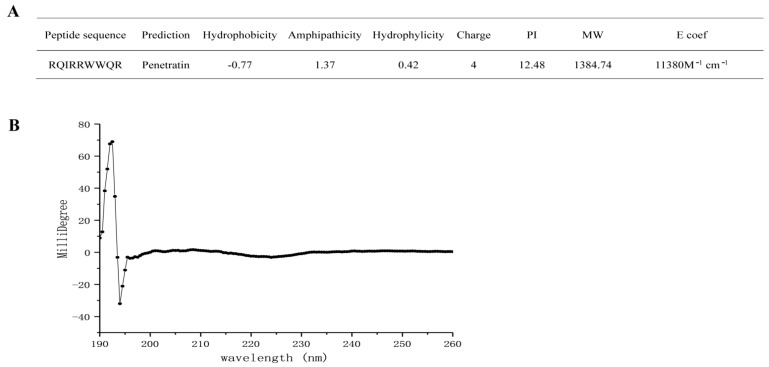
Molecular characteristics of RR9 (**A**) and CD spectra of RR9 dissolved in PBS at 37 °C (**B**).

**Figure 2 materials-14-02730-f002:**
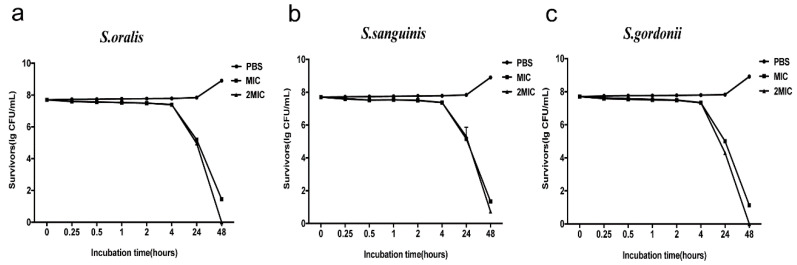
Antimicrobial activity kinetics of RR9 against three single species ((**a**) *S. oralis*; (**b**) *S. sanguinis*; and (**c**) *S. gordonii* ). Bacteria treated with RR9 were cultured for 48 h.

**Figure 3 materials-14-02730-f003:**
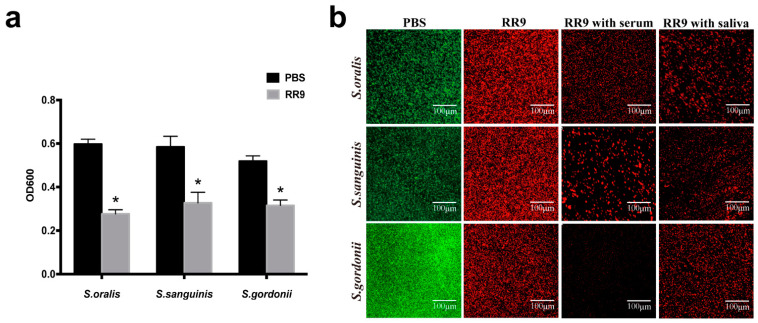
Antibiofilm effects of RR9 against single species (*S. oralis*, *S. gordonii*, or *S. sanguinis*) (**a**). The biofilms treated with RR9 (MIC = 25 μg mL^−1^) were incubated for 24 h. Data are shown as the mean ± SD; *n* = 3. * *p* < 0.01 compared with the control groups. Two-dimensional CLSM images after treatment with RR9 (25 μg mL^−1^), RR9 and serum, or RR9 and saliva against *S. oralis*, *S. gordonii*, and *S. sanguinis* (**b**). The overlay images show dead cells (red) and living cells (green), and the scale bar is 100 μm.

**Figure 4 materials-14-02730-f004:**
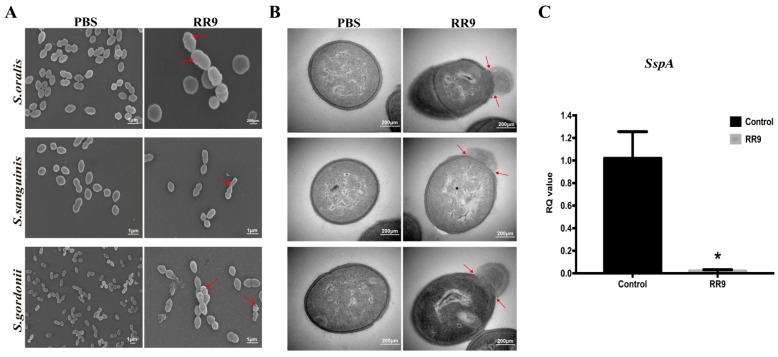
SEM images of *S. oralis*, *S. gordonii*, and *S. sanguinis* biofilms treated with RR9 at 25 μg mL^−1^ (MICs) for 24 h (**A**). TEM images of the bacteria treated with RR9 at 25 μg mL^−1^ for 12 h (**B**). Red arrows: released cellular contents and disrupted cell membrane. (**C**) Gene expression of *sspA* in *S. gordonii* treated with RR9 or left untreated. *S. gordonii* planktonic cells (1 × 10^8^ CFU mL^−1^) and cells treated with RR9 overnight at 25 μg mL^−1^ (MICs) for 24 h were harvested separately for RT-PCR. * *p* < 0.05.

**Figure 5 materials-14-02730-f005:**
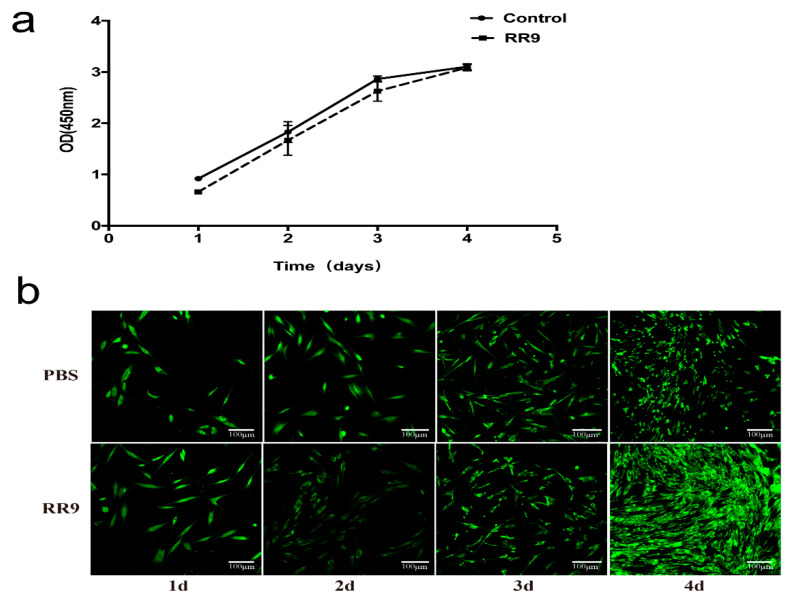
Proliferation curve of HGF cells treated with RR9 for 1–4 days and without RR9 treatment (control) (**a**). HBFs cells stained with AO/EB (green: live; red: dead) after 1–4 days of incubation in the absence (control) and presence of RR9 at 25 μg mL^−1^ (**b**). Data are shown as the mean ± SEM; *n* = 3.

**Table 1 materials-14-02730-t001:** The respective MIC and MBC values of RR9 against *S. oralis*, *S. gordonii*, and *S. sanguinis*.

Bacteria	MIC (μg mL^−1^)	MBC (μg mL^−1^)
*S. oralis*	25	62.5
*S. sanguinis*	25	62.5
*S. gordonii*	25	50

## Data Availability

All data included in this study are available upon request through contact with the corresponding author.

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
