# Peer review of "Antibacterial Properties of Small-Size Peptide Derived from Penetratin against Oral Streptococci"

_materials, 2021, doi:10.3390/ma14112730_

Round 1

Reviewer 1 Report

The study is well presented and meets the requirement of scientific soundness.

However,

  • Rows 268 - 269: the authors state : “and the connection be- 268 tween cells was more obvious (Fig. 5a) in 4 and 5 days.”

Observation:

- Figure 5a presents a graph that doesn’t include a curve from 4th to 5th day.

- Also, the description of Fig. 5 (a, b) states 1–7 days of incubation, but only days 1-4 are presented.

- Moreover, the type of analyses for Fig. 5b must be included in description.

Reviewer 2 Report

The aim of the article entitled „ Antibacterial properties of small-size peptide derived from penetratin against oral streptococci” was to determine the efficacy of the RR9 peptide against planktonic and biofilm forms of selected streptococci. The manuscript is multi-threaded and graphically well implemented (figures and charts). Additionally, multiple research techniques are also improving the perception of the article. I only have some reservations about the genetic part of the research (detailed description below).

Below I present a few minor adjustments, the inclusion of which will improve the quality of the article:

  • “… planktonic state and during biofilm formation in vitro with little toxic” -> planktonic state and during biofilm formation in vitro while keeping a low toxicity against eukaryotic cells. [line 24]
  • “The antibacterial mechanism proved to relate to the sspA gene in streptococci.” -> The antibacterial mechanism proved to be related with the lower expression of sspA in streptococci [line 25]
  • “… complicated relationship with other systemic diseases” -> please list a few of them [line 37]
  • “Plaque biofilm is the …” -> I believe it is worth adding a paragraph here or extending this paragraph with 1-2 sentences describing the oral biofilm [line 37]
  • “there is badly in need of” -> an unfortunate phrase, please change [line 50]
  • “the development an understanding …” -> the development of what? Please add [line 51]
  • In the "Materials and Methods" section, paragraph titles are not bolded and there is a lack of spaces between them; please add
  • “which will not be repeated here” -> this phare in lines 81 and 91 are not needed, please delete
  • Figure 1a should be larger, because at present it is difficult to read the values ​​- according to the requirements of the journal, the tables can be stretched across the full width of the page
  • “The results indicated … against the three selected primary colonizers” -> The results indicated … against the three selected primary colonizers, i.e., gordonii, S. sanguinis, and S. oralis. [lines 184-185]
  • “… which showed that S. gordonii is more susceptible to RR9” -> … which showed that gordonii is more susceptible to RR9 (MIC and MBC counting for 25 and 50 μg mL-1, respectively). [line 187]
  • “Bacterial death occurred within 0.25 h upon RR9 addition …” -> Bacterial death occurred within 4 h upon RR9 addition … [line 200] (a statistically significant decrease is observed rather after 4 hours of incubation)
  • “with RR9 (25 μg mL-1) were incubated for 24 h” -> with RR9 (MIC = 25 μg mL-1) were incubated for 24 h [line 227]
  • “SEM images of oralis, S. gordonii, and S. sanguinis biofilms treated with RR9 at 25 μg mL-1 for 24 h” -> SEM images of S. oralis, S. gordonii, and S. sanguinis biofilms treated with RR9 at 25 μg mL-1 (MICs) for 24 h [line 254]
  • gordonii planktonic cells (1 × 108 CFU mL-1) and cells treated overnight with RR9 for 24 h were harvested for RT-PCR.” -> What was the RR9 concentration used? please specify [line 257]
  • I have reservations about point 3.5. and gene expression studies. How is the certainty that the reduction in expression of sspA is not a result of bacterial death, since MIC concentrations reduce viability after 24h incubation?
  • Figure 5A seems a little crushed (height), if possible please correct it
  • The part in lines 364-370 is not totally clear; I suggest rearranging the words to make the text easier to understand
  • “Due to the side effects of antibiotics, adjuvant drug therapy for periodontitis must have fewer adverse effects” -> Due to the side effects of antibiotics, the simultaneous use of low concentrations of drugs in adjuvant therapies may induce fewer adverse effects during the periodontitis treatment. [line 372]

Reviewer 3 Report

The use of a penetratin derivative to control early-stage colonizers is interesting. The data presented are interesting, however, some inaccuracies detract from the message and need to be corrected. Line 87: the sentence "[...] to make its final concentration to." needs to be finished and the concentrations indicated. Line 100: The MBC test needs to be explained. The 99.9% inhibition rate was obtained in which type of experiment? Figure 2: The quantification of survivors is given in log CFU/mL and at t = 0 there are almost 10^8 CFU/mL while in Materials and Methods, line 109, says that it is 5 x 10^5 CFU/mL that are diluted 2 times. At t = 0, one would expect a value closer to 10^5 CFU/mL. Please review the conditions under which this experiment was performed on the three streptococcal species. Figure 3b: The absence of signal in the S. gordonii biofilm after 24h incubation of RR9 in serum is questionable. Is it possible that the image was not processed in the same way as the others? Line 257: write 1 x 10^8 correctly Figure 5: Proliferation curves were not performed with HGFs but with a more resistant MC3T3-E1 cell line that is closer to osteoblasts than fibroblasts. Please specify if some experiments were performed with HGFs. If not, please modify the manuscript, especially Line 360.

Round 2

Reviewer 2 Report

I would like to thank the Authors for carefully incorporating all suggested corrections. I believe that the quality of the manuscript has improved and is now suitable for publication.

Reviewer 3 Report

Thanks for your modifications